# Footwear Choice and Locomotor Health Throughout the Life Course: A Critical Review

**DOI:** 10.3390/healthcare13050527

**Published:** 2025-02-28

**Authors:** Kristiaan D’Août, Omar Elnaggar, Lyndon Mason, Adam Rowlatt, Catherine Willems

**Affiliations:** 1Department of Musculoskeletal and Ageing Science, Institute of Life Course and Medical Sciences, University of Liverpool, Liverpool L7 8TX, UK; omar777@liverpool.ac.uk (O.E.); a.rowlatt@liverpool.ac.uk (A.R.); 2Liverpool University Hospital NHS Foundation Trust, Liverpool L7 8YE, UK; lwmason@liverpool.ac.uk; 3School of Medicine, Institute of Life Course and Medical Sciences, University of Liverpool, Liverpool L7 8TX, UK; 4Department of Design, KASK and Conservatorium School of Arts, HOGENT, 9000 Gent, Belgium; catherine.willems@hogent.be

**Keywords:** aging, biomechanics, foot, footwear, health, heel pad, shoes

## Abstract

**Background/objectives:** In this paper, we review and discuss epidemiological and experimental evidence on the effects of daily footwear on gait and life-long health. We consider different types of footwear, including “minimal shoes”, and their design features, comparing them to barefoot walking, with a focus on overall gait and the function of the heel pad. **Methods:** Narrative review. **Results:** We find little evidence for health benefits of most typical shoe design features (e.g., cushioning, raised heels or arch support) for normal walking in healthy individuals, and in several cases (e.g., high heels), there is evidence of detrimental health effects. **Conclusions:** Based on currently available evidence, we recommend minimal footwear as the default for the general population to stimulate healthy biomechanical aging, with other types of footwear used infrequently or when there is an individual or clinical need.

## 1. Introduction

Humans are the only fully bipedal primates and have been bipedal for up to 6–7 million years [1], relying on their feet as the mechanical interface with the substrate. Because locomotion has played a crucial role during our evolutionary history (be it to collect food, escape predators, etc.), it is safe to assume that selective pressures on the foot and gait must have been great and have shaped the evolution of the foot. During most of the course of our evolution and into anatomical modernity, we performed all activities barefoot [2,3,4]. It is only relatively recently, approx. thirty thousand years ago [5], and well after we were anatomically modern, that we started to wear shoes. Footwear serves as an interface between the foot and the substrate, altering its biomechanics and potentially influencing gait patterns. This may be especially true for the heel area, which experiences the greatest forces and where anatomical cushioning will interact with any footwear cushioning, if present. In this critical review, we will explore what is currently known about these effects and what they mean for walking in a healthy population. However, we will also discuss running and abnormal gait where relevant. In many of the latter cases, footwear can be seen as an intervention or a performance-enhancing device rather than a mere item of clothing.

Before we discuss footwear, it is important to briefly consider barefoot walking. As mentioned, barefoot walking is the evolutionary normal condition for all hominins, including modern humans, as it is for any other animal species. Therefore, we consider barefoot walking as the starting point throughout this paper, and we have successfully performed without shoes throughout our evolutionary history. Similarly, we have not used, to date, other items of clothing that affect us biomechanically—for example, we do not wear supportive head collars unless there is a medical reason (or in some cases, a cultural reason) to do so. However, this starting point does not imply that barefoot walking is necessarily the “best” in terms of performance and health; this would be a naturalistic fallacy [6]. Throughout our history, and at an increasing pace, humans have excelled at “unnatural” inventions that have greatly helped us. These inventions range from stone tools to modern medicine, including footwear-based interventions, e.g., in diabetes [7]. In principle, it is possible that daily footwear is one such invention. Whether or not this is the case can only be answered by evidence, which we will attempt to do in this paper. This paper will consider two types of evidence: epidemiological (population health effects of shoes) and experimental (usually lab-based functional analyses).

Why do we wear shoes? Seemingly trivial reasons are protection from the cold, infections, and traumatic injury. However, in some cases, the effects of these factors may not be as straightforward as assumed. For example, local habituation of the feet to the cold develops relatively easily [8], and naturally developed calluses offer mechanical protection [9,10]. There are also social and cultural reasons to wear shoes. It is important to note that these reasons do not necessitate the shoes to have a mechanical function, and indeed, several types of indigenous footwear lack mechanical functions and can be considered “minimal footwear” [11]. Minimal footwear is footwear that aims at replicating barefoot walking and is typically flexible, foot-shaped, and without features such as a substantial amount of cushioning, arch support, or a raised heel [12]. Most types of conventional footwear, however, have such features, and in this paper, we will discuss their effect on gait and health.

The overall aim of this paper is to evaluate current evidence, stimulate a discussion on the optimal choice of footwear to stimulate long-term biomechanical health in the general population throughout aging, and make footwear selection a more deliberate health choice than is currently the case. Despite the prevalence of conventional shoes, their long-term biomechanical effects remain poorly understood. Key questions are as follows: How does footwear affect our lifelong health? What are the functional effects of the most common design features in conventional shoes?

## 2. Methods

The scope of this paper is around the influence of footwear on lifelong locomotor health, focusing on the general population. Because of this broad scope, we approach this paper as a critical review (one type of narrative review), as this is flexible, allows for a practical synthesis of a broad topic, and brings an interpretative perspective [13,14]. Our scope excludes clinical or sports footwear, but we have cited some examples of these fields where relevant for the general population. We included studies comparing habitually barefoot and shod populations, experimental studies, and epidemical research on foot health.

Our literature search strategy was as follows. We used PubMed, Google Scholar, and our personal academic library to search for publications, using keywords corresponding to the respective section of the paper (e.g., “cushioning”, “heel cup”, “arch support”). Since this is not a systematic review, additional search queries, as deemed relevant, were used, and we also consulted the reference lists of papers identified. We considered peer-reviewed articles written in English, including both original studies and reviews, published any time up to and including November 2024.

The content of this paper is subdivided into two main sections. The first (Section 3) deals with the effect of footwear, in general, on health. The second (Section 4) deals with common design features of shoes and their effects on foot anatomy, gait, and health.

## 3. Foot Health and Footwear

The interaction between footwear, biomechanical function, and health is complex and shaped by various factors ranging from individual foot anatomy to the specific features of different types of shoes. In this paper, we distinguish three footwear conditions: barefoot, minimally shod, and conventionally shod.

Barefoot: Unshod walking represents the ancestral state for the human foot. Despite this, most research on the human foot has been conducted within shod populations. Without any footwear, the foot interacts directly with the ground, allowing for unrestricted movement and the use of all foot muscles and structures. Barefoot walking also promotes the development of thicker calluses, which provide natural protection without sacrificing tactile sensitivity [9]. Additionally, barefoot individuals often exhibit a more natural arch and foot shape, as the foot is not constrained by shoe structures [15].

Minimally shod: So-called minimal footwear (sometimes the oxymoron “barefoot shoes” is used) aims to replicate the barefoot experience while offering some protection against environmental hazards. These shoes are typically characterized by a thin, flexible sole, a lack of arch support, and a foot-shaped design that allows for natural toe splay. The primary goal of minimal footwear is to enable natural foot movement and muscle activation, similar to walking barefoot. Research indicates that minimal footwear can help maintain natural gait patterns and reduce the risk of certain foot deformities associated with more restrictive shoes [12]. However, minimal shoes still offer some degree of protection from sharp objects and abrasive surfaces, making them a practical alternative for those seeking the benefits of barefoot walking in urban or rugged environments [11].

Conventionally shod: Conventional footwear includes most modern Western shoes, which are designed with various features aimed at comfort, support, and performance. These shoes typically have thicker soles, cushioning, arch support, and a raised heel. While these features may provide immediate comfort and support, especially on hard surfaces, they also alter natural foot mechanics. Additionally, arch support and cushioning can weaken intrinsic foot muscles over time (see below). Despite these potential drawbacks, conventional shoes may be beneficial in specific contexts, such as providing extra support for individuals with certain foot pathologies or during high-impact activities [7].

Aging affects foot biomechanics, and a variety of changes within the foot’s structure and biomechanics occur. This includes changes in the skin, affecting mechanical properties of the plantar surface, leading to greater hardness and foot problems [16,17]. In addition, the range of motion is reduced within the ankle and metatarsals with decreased internal strength within the muscles of the foot. As a result, older people generally develop a more pronated posture during gait [17]. Older people have been shown to be less efficient with slower walking speeds and reduced step lengths, resulting in a reduction in forces experienced during locomotion and longer contact durations [18]. Because of age-related changes, optimal footwear may change along the life course.

We will explore the effects of footwear on the foot, gait, and health by focusing on three lines of evidence. The first is a comparison between habitually barefoot versus shod adult populations, which can provide an insight into (life) long-term effects. The second is an assessment of how footwear impacts children, whose feet are growing and ossifying; thus, the impact of footwear may be especially visible and have long-term effects. The third focuses on the here-and-now effects of different footwear conditions on gait, i.e., how the barefoot or minimally shod condition affects gait in otherwise habitually shod adults.

### 3.1. Long-Term Effects of Footwear: Comparing Habitually Barefoot and Habitually Shod Populations

In this section, we will discuss footwear in general (i.e., compared to barefoot) and its long-term impact on the foot and general health. Unfortunately, interventional studies on the use of footwear over a lifetime or at least several year periods are non-existent. Our best alternative is to rely on epidemiological data to compare gait and foot health between habitually barefoot and shod walkers. There are inherent limitations to epidemiological studies, i.e., they are descriptive, and confounding factors may (partly or entirely) explain observed differences between shod and unshod populations. For example, Zech et al. [19] found that barefoot (South African) children had better balance but lower sprint speed than shod (German) children. Shu et al. [20] found anatomical differences between shod (Chinese) and unshod (Indian) runners [21]. Whilst of epidemiological interest, such studies cannot causally link the observed differences to footwear factors. Indeed, foot differences have been reported between different populations (e.g., [22,23,24,25,26]), and we need to include more diverse populations in our studies than is the case to date. Other confounding factors include socioeconomic status, substrate hardness, and cultural differences. It should be noted that direct measures of overall health are rarely available, and long-term health effects are difficult to assess [27]. Most studies rely on proxy measurements or foot-specific features such as arch structure, foot deformities (e.g., hallux valgus), or gait performance.

The most informative studies are on populations that are partly or fully habitually barefoot or on children. Indeed, whilst the adult foot is fully ossified and has limited scope to change its shape (and function), the infant foot is not ossified until the age of ten, is highly plastic, and therefore will reflect the effect of footwear much more clearly than is the case in adults [28,29]. In this section, we will discuss adult barefoot populations, and in the next section, we will discuss children.

There are several studies directly addressing barefoot populations. Some of these studies are historical and may contain language or perspectives that are outdated by current standards, but their findings are considered informative and relevant to contemporary footwear choices.

Hoffmann [30] studied foot shape in barefoot populations from the Philippines, Central Africa (as well as the feet of ancient Greeks as shown in statues), and a Western shod sample. He found that adult forefeet are narrower in shod populations than in barefoot populations and that this is a result of footwear because infant feet are largely similar across populations. He also found that pathological flat feet hardly exist in barefoot populations. James [31], studying 65 habitually barefoot Solomon Island inhabitants, found that all had well-developed arches, which he attributed to the lack of footwear. Shulman [32] studied over 5000 habitually barefoot people from China and India. He found that foot defects were very rare (9% for all conditions studied) and non-debilitating, and he suggested that “shoes are not necessary for healthy feet and are the cause of most foot troubles”. Arch height was lower than in the US but did not lead to functional impairment. Sim-Fook and Hodgson [33] did a within-population (China) comparison between 118 habitually barefoot and 107 shod people. They found that the unshod foot had thicker, protective skin, a wider forefoot, and a much lower incidence of hallux valgus (1.9% vs. 33%), and, in line with Shulman [32], cases of flat foot were rare and asymptomatic. Kadambande et al. [34] found that habitually barefoot walkers had more pliable feet. Ichikawa et al. [35] compared lifesavers, who spend a lot of time barefoot on sand, to the general population and found that they had higher arches and increased foot muscle mass. Stolwijk et al. [36] found that foot complaints are rare in Africa, with people from Malawi having lower arches, different dynamics, and more equal pressure distribution (as in [37] for an Indian population). Please note that foot shape and size can differ between populations, regardless of the effect of footwear. The studies mentioned here suggest that habitually barefoot people have healthy feet with well-developed arches and low incidence of foot pathologies.

### 3.2. Long-Term Effects of Footwear: Foot Development in Children

Several studies have addressed the development of children’s feet, with a strong focus on the medial longitudinal arch. For comprehensive reviews of the influence of footwear on children’s gait and foot development in children, we refer to Wegener et al. [38] and Squibb et al. [39], and we will present some key findings from habitually barefoot and shod children below.

Several studies address Indian children. A large-scale study on 2300 children found that the habitual use of footwear is associated with a higher incidence of flat foot [40]. Vangara et al. [41] found that tribal, habitually barefoot children had a higher incidence of high-arch feet than flat feet, with the latter decreasing with age from about 62% at age 3–4 to about 23% at age 14–15. A comparison between rural (mostly barefoot) and urban (mostly shod) children suggested that barefoot walking favours good arch development and may prevent overuse injuries [42]. A large (*n* = 1846) retrospective study found that participants starting to wear shoes later in their development (age 16) were about half as likely to have flat feet than those who started wearing shoes at a younger age [43].

Studies from Africa show that habitually barefoot Kenyan children had much lower injury rates than their shod peers [44]. Hollander et al. [15] studied barefoot South African children (although they might have worn constricting shoes infrequently) and compared them to German (shod) children. They found that the South African children had a higher arch, among other differences, but the authors did not draw conclusions on long-term health implications. A Nigerian study on 990 school children found that early introduction of shoes may predispose to flat foot [45]. Echarri and Forriol [46] studied a large sample (*n* = 1851) of urban (shod) and rural (barefoot) Congolese children and concluded that flat foot, common at a young age, tends to disappear with age in both shod and unshod populations, with a small effect of footwear as such.

Combined, these studies on children strongly suggest that barefoot walking provides the best conditions for the healthy development of the foot, in line with current recommendations on children’s footwear [47,48,49,50,51,52].

### 3.3. Short-Term Effects of Footwear: Studies on Barefoot, Minimally Shod, and Conventionally Shod Walking in Habitually Shod Populations

There are mechanistic studies that either address the effect of footwear on walking cross-sectionally or by using a longitudinal design (usually a few months in duration). Barefoot and minimally shod running has been well studied [27,53,54,55,56,57,58] but falls outside the scope of this paper. Such experimental studies allow the establishment of causal relationships between footwear and their effect on gait and feet. Here, we focus on footwear in general compared to barefoot walking or minimally shod walking but not specific features of footwear. Walking in minimal shoes resembles, but is not identical to, walking barefoot and involves different motor strategies [59]. This is likely because even minimal shoes may have some mechanical function and affect mechanoreceptor sensitivity.

Shod and barefoot walking in habitually shod populations differs in several respects. In terms of spatiotemporal gait variables, shod walking involves longer stride lengths for a given speed (and hence, a lower stride frequency), a greater duty factor, and a faster preferred speed. In terms of joint kinematics, shod walking involves a more dorsiflexed ankle at heel strike, a more extended knee during heel strike, less dynamic spreading of the loaded forefoot, and smaller medial longitudinal arch motion [60,61,62]. Minimal and barefoot walking leads to lower muscle activation in some leg muscles [63,64], except possibly in the gastrocnemius [63], which, as a plantar flexor, is largely responsible for powering walking [65,66].

Using minimal footwear has acute effects on gait parameters, e.g., walking with shorter strides but at a higher cadence [67], as in barefoot walking, and leads to acute improvement in static and dynamic stability ([68,69,70], contra [71]) and spatiotemporal gait parameters [72]. In the long term, when adopted as daily footwear, minimal footwear will increase foot strength and arch stiffness by approximately 60% [73,74] and improve balance [75]. Minimally shod walking is generally found not to be identical to barefoot walking; it can often be seen as intermediate between barefoot and conventionally shod walking [63] and, in older people, can improve gait performance [76] and balance [70] more than barefoot walking (reviewed in [77]).

Even though we focus on the general population in this paper, it is noteworthy that some of the benefits of minimal or barefoot walking can be reflected in patient populations, too. For example, and perhaps counter-intuitively, knee loading is decreased in osteo-arthritis patients when walking barefoot or wearing minimal shoes compared to conventional shoes [78,79,80]. In users of (instrumented) prosthetic hips, barefoot and minimally shod walking reduced joint loading compared to conventionally shod walking [81]. However, at least in running, it has been shown that a relatively fast transition from conventional footwear can induce bone marrow edema [82], and it is safe to assume that, also for walking, a transition to minimally shod or barefoot locomotion needs to be gradual.

## 4. Effects of Specific Footwear Features on the Structure and Function of the Foot

Based on the studies outlined above, we assume that barefoot walking, or walking in “minimal” footwear that has no biomechanical function (but can offer trauma or thermal protection), should be the norm for non-clinical populations. With this as the starting point, which additional footwear features can we use to further improve footwear? Here, we will consider the most common features of footwear, as depicted in Figure 1, that likely have a mechanical effect.

### 4.1. Raised Heel

Several studies have addressed the effect of heel height (usually quantified as heel-to-toe drop) on gait. They found that there is a graded response, with high heels affecting the kinematics and kinetics of gait and muscle activities [83,84,85]. The raised heel shifts the body’s centre of gravity forward, which can affect posture and gait [86]. Achilles tendon loading is increased when a heel is used compared to barefoot, but because of increased changes in ankle configuration at greater heel heights, tendon loading is reduced again at higher heel heights [87]. It should be mentioned that in certain musculoskeletal pathologies, such as Achilles tendinopathy, a raised heel is successfully used to temporarily offload the Achilles tendon [88,89]. In any case, there is better scope for offloading if the triceps surae muscle-tendon unit (incl. the Achilles tendon) has not been already shortened [90] by habitually wearing heels.

Heeled shoes were historically used by all genders for a variety of practical (e.g., keeping feet clean on a dirty substrate, horse riding) and social status-signalling reasons [91]. Today, high heels may have psychosexual benefits [92,93] but are biomechanically unhealthy. Moderate heel heights are less problematic than higher heel heights, but we did not find any evidence that they would be beneficial for the general population.

Heeled gait leads to a less fluent gait pattern with more plantar flexed ankles and more flexed knees with larger accelerations, increased knee extensor and ankle plantar flexor activity, but its effect is reflected further up the body as well [83]. Interestingly, the latter does not lead to a greater plantar flexion moment due to the reduction in Achilles tendon moment arm as a result of the increased ankle plantarflexion [94]. Heels lead to an increased metabolic cost of walking [94,95], probably caused by a shift in work output from the ankle in normal walking to the hip and knee in heeled walking [96]. Anatomically, the reported increased plantarflexion leads to structural shortening of the calf muscle-tendon unit [90]; therefore, wearing heels may become a necessity once one is used to it.

Wearing high heels is associated with increased joint stress, possible patellofemoral pain [97], foot pain [98,99], the development of valgus deformity of the hallux and fifth toe [100,101,102], flatter transverse arches [102], increased plantar pressure [103], and increased risk of traumatic injury [84,104].

### 4.2. Cushioning

The provision of cushioning is typically seen as one of the key features of footwear. Because we have an anatomical, compressible heel pad arranged in series with any footwear cushion, and because the heel pad changes with aging, we will briefly discuss this before addressing footwear cushioning.

Changes with aging in soft tissue properties in general have been suggested to be responsible for the increase in foot problems such as heel pain and ulcerations [105]. Because of the high forces on a small area experienced during heel strike, the heel area deserves special attention. The heel pad is a structure between the calcaneus (heel bone) and the plantar skin, and it consists of fibroelastic septae with fat globules and significant nerve and blood supply [106]. It plays an important role in gait [107]; is associated with common foot issues such as plantar heel pain, plantar fasciitis, and Achilles tendonitis [108,109,110]; and is important for dissipating impact peaks during heel strike in healthy gait [111]. With age, the heel pad becomes stiffer and loses some of its shock-absorbing qualities [105,110,112]. The underlying mechanism for this is unclear but has been suggested to include accumulated mechanical loading (“wear and tear”) over the life course. However, heel pad thickness does not differ between active and sedentary individuals [113], which suggests that the degradation may be an effect of healthy aging itself. In addition to aging, several other factors have been shown to affect the heel pad. These include sex, with thicker heel pads seen in males than in females [114,115], and diseases such as diabetes ([116,117,118,119]). Ethnic background seems to have an influence, too; studies from Africa show that their participants have a thicker heel pad compared to Western populations [120,121,122,123]. It has been reported that habitual barefoot walkers have thicker heel pads than shod peers [121], and a simulation of barefoot vs. shod loading conditions reveals a different stress distribution on the heel pad [124], which may, in turn, lead to long-term changes.

Changes in the heel pad with aging (and other factors) are seen in its histology, thickness, and mechanical properties. Histologically, degraded heel pads exhibit less well-organized septae [106], a breakdown of collagen and elastin, and shrunken fat globules [125,126]. Studies on heel pad thickness with aging produce conflicting results. While most studies report thinning with age [121,127,128,129], some studies did not find a change [123] or found an increase in thickness [112,130]. In terms of dynamic mechanical properties, the heel pad is a visco-elastic structure, which has been characterized in vitro and in vivo. In vitro techniques include impact testing [111], whilst in vivo techniques include loaded/unloaded ultrasound [113,131], compression testing [132], and biplanar X-ray imaging with pressure recordings [127]. The outcomes include deformation, energy absorption [110], and stiffness [112]. Collectively, studies using these methods show that the heel pad is typically compromised as a shock absorber during aging and in disease [107], although there is some disagreement regarding whether the heel pad is less stiff with more energy absorbed [112] or more stiff with less energy absorbed [110] in older people. The older heel pad recovers more slowly after the removal of each load [105,127,133,134,135]. Modelling suggests that a healthy heel pad needs a compromise stiffness at which two health-related but conflicting responses, force loading and deformation, are at medium levels [136].

Most studies on footwear cushioning focus on running rather than walking because the impact (measured as, e.g., peak ground reaction force, loading rate, or tibial acceleration) during running is much higher, and overload injuries are very common [137]. Counter-intuitively, peak impact is unaffected by midsole hardness in running ([138] contra [139]), which might be explained by the observation that the heel pad compresses only half when shod compared to barefoot [140]. The impact loading rate can be even lower barefoot when using a forefoot or midfoot strike rather than a heel strike [54]. Overall, the effect of footwear cushioning on impact attenuation is complex (see [141] for a review) and merits more research. Importantly, a recent Cochrane Review on running shoes found no evidence that the shoe type (incl. minimal, soft, stable, and “motion control” shoes) affects the lower limb injury rate [142].

Barefoot walking (as opposed to barefoot running) does not involve a switch to a forefoot strike, but does lead to a more plantarflexed foot during heel strike, which offsets the lack of cushioning, and impact accelerations remain overall quite similar [11]. A “lighter tread” with lower impact forces (but higher loading rates) was also observed in barefoot walking compared to walking in minimal sandals [143].

Compared to running, the magnitude of foot strike impact (measured as, e.g., force, loading rate, or deceleration) is relatively low in walking and is not necessarily bad. Indeed, loading is needed for healthy bone formation, and it is argued that even in people with osteoporosis, it should not be completely avoided for this reason [144]. However, we do not know the optimal amount of impact loading.

We see no evidence of health benefits to incorporate a substantial amount of cushioning in daily footwear for mechanical reasons; in addition, cushioned shoes reduce stability [69,70] and somatosensory perception [145,146]. However, we see at least two possible reasons why a small amount of cushioning might be beneficial (but this requires more research), especially for older people.

The first reason is that we overwhelmingly live on hard and uniform artificial substrates [147,148], which has opened suggestions to incorporate barefoot surfaces in urban design [149], but it should be noted that some natural substrates are hard as well [11]. The main difference between artificial and natural substrates might be the amount of variation in topology and mechanics rather than hardness per se.

The second reason is that, as mentioned previously, the damping qualities of the heel pad decline with age [110], and a small amount of cushioning might offset that functional loss. We suggest that more research is needed to assess if a small amount of cushioning may be beneficial or at least not detrimental (and offer subjective comfort). It should be mentioned that from a purely mechanical point of view, the amount of cushioning needed to substantially reduce impact on a hard substrate (by extending the duration of the impact) can be small and possibly as little as that provided by a few mm of leather or rubber.

### 4.3. Arch Support

Humans, unlike other primates, have foot arches, with the medial longitudinal arch being the most developed. This arch develops naturally during childhood [48]; this seems to happen across different types of footwear [150], but it may be the most noticeable when walking barefoot (see above). The foot arches are supported by the bony architecture, ligaments, and muscles of the foot. Longitudinal arch height varies between individuals and populations, and it is not clear what a “good” arch height is. Even though a low arch (“flat foot”) can be pathological [151], in many cases, it is not. However, at least some flat feet are pathological. Interestingly, in a barefoot Indian population, the average arch height is somewhat low but with a small variation, so the incidence of flat feet is lower than in some populations with a greater average arch height [37]. Athletes with flat feet may be at no additional injury risk [152], and in some populations, low arches are normal [36]; in general, there is no evidence that asymptomatic flat feet will develop to become symptomatic [153]. Unlike the heel, metatarsal, and toe areas of the foot, the medial midfoot area has thin skin and no damping structures like a heel pad and is, therefore, less suited to be directly loaded plantarly. Notwithstanding, “arch support” is a common feature in daily footwear.

There are specific situations where arch support may be beneficial. In people with flat feet and fatigue, arch support can reduce pain and decrease energy consumption during walking [154]. However, arch support can also lead to changes in spatiotemporal gait parameters [155]. Arch supporting orthotics may help correct pathological conditions even though the evidence is often weak or anecdotal [156,157]. An in vitro study has shown that arch support can help offload plantar aponeurosis [158].

For the general population, we did not find empirical evidence that external arch support provides health benefits, and if we can extrapolate the studies on arch development in children wearing shoes, which in many cases will have built-in arch support, then arch support might lead to poorer arch development.

### 4.4. Last Shape and Shoe Size

It has long been acknowledged that most shoes, traditionally constructed using rigid lasts, do not accurately represent the shape of the human foot. This misalignment is particularly evident in the toe box area, which is often too narrow, failing to accommodate the natural spread of the toe area [30,159]. Studies have consistently shown that ill-fitting shoes are a widespread issue: 88% of women wear shoes that are narrower than their feet by an average of 1.2 cm [160,161], and 90% of a mixed-gender sample of older adults wear shoes smaller than their feet [162]. Among children, approximately 80% wear shoes that are too small, which is linked to the development of conditions like bunions and hallux valgus [163].

Further compounding the issue, wearing shoes that are too small—either in length or width—or improperly shaped can lead to a variety of deformities, including bunions, poorly developed arches, and hallux valgus [101,163]. Such footwear also frequently results in pain [164,165]. When walking barefoot, the width of the children’s foot increases by 9.7% across the stance phase, a natural expansion that is restricted to 4-6% when wearing shoes, thereby inhibiting the foot’s natural movement pattern [52].

It is crucial for footwear design to not only respect both the static and dynamic aspects of the foot’s shape [166] but also to ensure that shoes are not overly large, which could diminish toe clearance and increase the risk of tripping and falling [167]. Finding this balance is essential to promote foot health and prevent mobility issues. In addition, it must be recognized that foot shape and size differs between populations and sexes [168], and lasts need to be designed accordingly.

### 4.5. Bending and Torsional Stiffness

Shoe stiffness has been primarily studied in athletic footwear, where a high bending or torsional rigidity is often seen as a performance-enhancing feature, but its use in sports is controversial [169,170]. In some pathologies, a stiff shoe can be an important part of treatment (e.g., the diabetic foot [171]).

For daily footwear and walking, we are not aware of evidence that adding bending or torsional stiffness has health benefits, and archeological evidence suggests that stiff shoes may produce narrower feet and a higher metatarsal torsion [172].

### 4.6. Rigid Heel Counter

Many shoes have a rigid, “supportive” heel counter, suggested to help control rearfoot movement such as pronation [173,174]. Heel counters extend medially and laterally to the longitudinal arch [175]. They are designed to stabilize the heel and control motion, but their effects can be multifaceted. For instance, while heel counters are thought to increase the absorption of forces during heel strike [115,176], a recent in vitro study showed that heel pad confinement actually makes the heel pad behave stiffer [177]. This increased stiffness could influence not only the heel but also the forefoot, with heel counters potentially increasing the forces experienced during movements such as jumping and landing within the forefoot [178]. Although jumping and landing are not typical of walking gait, this raises the question of whether heel counters might also influence the forefoot across the stance phase.

The studies mentioned here focus on conventional footwear, and it can be questioned whether a heel counter would be beneficial with minimal shoes, which, like barefoot walking, may require less motion control. The biological heel is rounded and produces a relatively small moment arm of the ground reaction force at heel strike compared to a shoe with a stiff sole that extends beyond the anatomical heel. Therefore, barefoot or minimally shod footwear is expected to produce smaller moments (and probably motions) even at similar or possibly higher forces.

The above section of footwear features has highlighted that conventional Western shoes incorporate design features that lack proven health benefits and, in some cases (e.g., heels), are correlated with health risks. The existence of these features can likely be best explained in a historical context. It is noteworthy that many types of indigenous footwear do not possess these features and indeed resemble minimal shoes [179].

## 5. Conclusions

A critical review of the existing literature suggests that habitual barefoot walkers have relatively healthy feet and that barefoot walking helps normal foot development in children. Conventional Western shoes offer protection against the elements and infections, but most of their design features have no proven biomechanical benefits and can be detrimental.

For the general population, we see no arguments in favour of shoes interfering mechanically with the body, but this is what many design features do. We propose to extend the medical principle “do no harm” to footwear for daily life by proposing the use of minimal shoes (with minimal mechanical interference with gait) as the default for the general population.

Shoe design features that do affect biomechanical function should only be incorporated when they have proven health benefits. This can be the case in the clinic, when a shoe becomes an intervention rather than a clothing item, or in sports, where a shoe becomes a piece of equipment, but is not clear for day-to-day footwear for the general population. We suggest that the use of minimal shoes has the potential to improve life-long locomotor health in the general population.

Further, ideally longitudinal, research is needed to determine the long-term effects of footwear types across diverse populations and varying environmental conditions.

## Figures and Tables

**Figure 1 healthcare-13-00527-f001:**
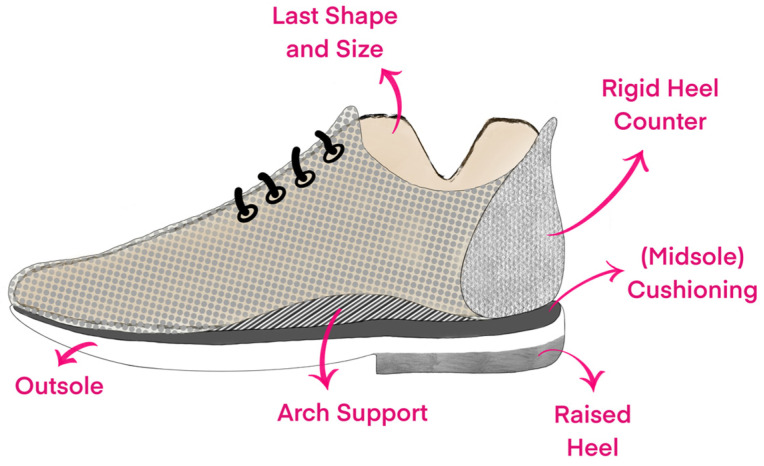
Graphical illustration of the common design features in conventional footwear.

## Data Availability

Not applicable.

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
