# Peer review of "Footwear Choice and Locomotor Health Throughout the Life Course: A Critical Review"

_healthcare, 2025, doi:10.3390/healthcare13050527_

Round 1

Reviewer 1 Report (Previous Reviewer 3)

Comments and Suggestions for Authors

To the author

Congratulations on a great paper.

Here are a few suggestions for improvement

1. in the methodology, you should add the period of time you searched for papers for the keywords you wanted to find. For example, you should add that you searched for papers from January 2000 to January 2025 and reviewed some of them.

2. In the results, the content of the reviewed papers for each topic is written in a narrative, but I think it would be a better paper if it was organised in a more systematic table.

3. in the summary and conclusion, you need to add a discussion section by citing the papers that were not available in the paper database.

Anyway, have a good day.

Author Response

Reviewer 1

*** Comment 1 ***

Congratulations on a great paper.

Here are a few suggestions for improvement

  1. in the methodology, you should add the period of time you searched for papers for the keywords you wanted to find. For example, you should add that you searched for papers from January 2000 to January 2025 and reviewed some of them.

*** Response 1 ***

Thank you, the manuscript clarifies that any papers up to November 2024 (including) were considered.

*** Comment 2 ***

  1. In the results, the content of the reviewed papers for each topic is written in a narrative, but I think it would be a better paper if it was organised in a more systematic table.

*** Response 2 ***

Thank you, and we used to have a table in a previous version of the manuscript (along with text). However, upon a reviewer’s request we took it out. We believe the narrative is sufficient to communicate our message so prefer not to put tables back in.

*** Comment 3 ***

  1. in the summary and conclusion, you need to add a discussion section by citing the papers that were not available in the paper database.

Anyway, have a good day.

*** Response 3***

Thank you, but we are not sure what is meant by “citing papers that were not available”. All papers we cited are available to us in full format. If any potentially relevant papers would not be available we wouldn’t be able to cite them.

Reviewer 2 Report (New Reviewer)

Comments and Suggestions for Authors

   The manuscript presents a critical review on the impact of footwear on locomotor health across the lifespan. The review suggests that minimalist footwear should be the preferred choice for the general population, arguing that conventional footwear lacks proven biomechanical benefits and may have detrimental effects. While the manuscript covers an important and relevant topic, several areas require improvement in terms of clarity, scientific rigor, organization, and methodology. Below are detailed suggestions for revision, categorized by section.

1. Abstract & Simple Summary

Issues:

  • The abstract lacks clarity and coherence, especially in summarizing key findings.
  • Some sentences are incomplete or grammatically incorrect.
  • The key takeaway message is not clearly defined.

Specific Recommendations:

  1. Clarify the main objective.

    • Current wording:
      • “Here we, gait and health. However, footwear selection rarely takes this into account.” (lines 13-15) – This sentence is incomplete and unclear.
    • Suggested revision:
      • “Here, we examine the impact of different types of footwear on gait and health. However, footwear selection rarely considers these biomechanical implications.”
  2. Improve sentence structure and clarity.

    • Current wording:
      • “Our review suggest that minimal footwear, which interacts minimally with the way we walk, is preferable for healthy peopleto stimulate long-term health in the general pop-ulation.” (lines 17-19) – Contains grammatical errors.
    • Suggested revision:
      • “Our review suggests that minimal footwear, which minimally interferes with natural gait, may be preferable for promoting long-term musculoskeletal health in the general population.”
  3. Abstract should emphasize findings rather than general statements.

    • Consider adding specific findings from the review, such as the effect of footwear on muscle activation, joint loading, and foot morphology.

2. Introduction

Issues:

  • The introduction is well-grounded in evolutionary context, but it does not clearly articulate the research gap.
  • Some sentences are redundant or overly complex.
  • The transition from barefoot walking history to modern footwear issues is abrupt.

Specific Recommendations:

  1. Improve clarity in the opening statement.

    • Current wording:
      • “Humans are the only bipedal primates and have been bipedal for up to 6-7 million years [1]. This means that our feet are the only mechanical interface with the substrate.” (lines 32-34)
    • Suggested revision:
      • “Humans are the only fully bipedal primates, relying on their feet as the primary mechanical interface with the ground. Bipedalism has shaped the evolutionary development of the foot’s structure and function.”
  2. Strengthen the transition between historical context and modern footwear concerns.

    • Current wording:
      • “Because a shoe sits physically between the foot and the substrate, it will affect the biomechanics the foot and of overall gait to some degree.” (lines 40-41) – awkward phrasing.
    • Suggested revision:
      • “Modern footwear serves as an interface between the foot and the environment, altering foot biomechanics and potentially influencing gait patterns.”
  3. Clarify research gap and objectives.

    • The manuscript should explicitly state why this review is necessary.
    • Suggested addition:
      • “Despite the prevalence of conventional footwear, its long-term biomechanical effects remain poorly understood. This review aims to evaluate current evidence on how different types of footwear influence lifelong locomotor health.”

3. Methods

Issues:

  • Lack of systematic methodology. The paper does not clearly outline inclusion/exclusion criteria for studies.
  • No clear justification for database selection (Google Scholar and PubMed).
  • Potential selection bias as the methodology is not transparent.

Specific Recommendations:

  1. Define a structured search strategy.

    • Suggested revision (to be added in the methods section):
      • “A literature search was conducted using PubMed and Google Scholar up to November 2024. Keywords included ‘footwear biomechanics,’ ‘barefoot walking,’ ‘foot morphology,’ and ‘gait analysis.’ Only peer-reviewed English-language studies were included.”
  2. Explain study selection criteria.

    • The manuscript should state:
      • “We included studies comparing habitually barefoot and shod populations, experimental studies on footwear effects, and epidemiological research on foot health.”
  3. Clarify how the review was conducted.

    • Was it a narrative review or a semi-systematic review? The authors should specify their approach.

4. Results & Discussion

Issues:

  • Overgeneralized conclusions without sufficient nuance.
  • Limited discussion of confounding factors (e.g., socioeconomic status, surface hardness, cultural differences).
  • Some claims lack strong supporting evidence.

Specific Recommendations:

  1. Avoid absolute statements when evidence is mixed.

    • Current wording:
      • “Conventional footwear lacks biomechanical benefits.” (lines 495-496)
    • Suggested revision:
      • “While some design elements of conventional footwear, such as excessive cushioning and arch support, may alter natural foot mechanics, their long-term impact on health remains debated.”
  2. Include discussion on adaptation to footwear.

    • Some studies suggest that foot adaptations to footwear occur over time.
    • Suggested addition:
      • “Recent studies indicate that habitual use of footwear may lead to structural and functional adaptations in the foot, which should be considered when evaluating the effects of conventional vs. minimal footwear.”
  3. Acknowledge potential benefits of certain shoe features.

    • Raised heels can be useful in conditions such as Achilles tendinopathy.
    • Arch support may benefit individuals with flat feet experiencing pain.

5. Conclusion

Issues:

  • The conclusion is too strong given the available evidence.
  • The call to action lacks nuance—suggesting that all conventional footwear is harmful is an overgeneralization.

Specific Recommendations:

  1. Rephrase the concluding statement to reflect a more balanced perspective.

    • Current wording:
      • “Minimal footwear should be the default choice for the general population.” (lines 500-501)
    • Suggested revision:
      • “Based on current evidence, minimal footwear may offer biomechanical advantages for healthy individuals. However, specific footwear choices should consider individual needs, environmental conditions, and potential medical requirements.”
  2. Highlight knowledge gaps for future research.

    • Suggested addition:
      • “Further research is needed to determine the long-term effects of different footwear types across diverse populations and varying environmental conditions.”

Final Remarks

  • The manuscript covers an important and underexplored topic, but improvements in clarity, methodology, and balance of argumentation are necessary.
  • A clearer methodological framework and more nuanced discussion would greatly enhance its scientific impact.
  • Language polishing is recommended to improve readability and coherence.
Comments on the Quality of English Language

The manuscript is generally understandable; however, it contains numerous grammatical errors, awkward phrasing, and inconsistent terminology. Some sentences are unclear or incomplete, affecting readability. Additionally, word choice and sentence structure could be improved for better clarity and fluency.

Author Response

Reviewer 2

*** Comment 1 ***

The manuscript presents a critical review on the impact of footwear on locomotor health across the lifespan. The review suggests that minimalist footwear should be the preferred choice for the general population, arguing that conventional footwear lacks proven biomechanical benefits and may have detrimental effects. While the manuscript covers an important and relevant topic, several areas require improvement in terms of clarity, scientific rigor, organization, and methodology. Below are detailed suggestions for revision, categorized by section.

  1. Abstract & Simple Summary

Issues:

  • The abstract lacks clarity and coherence, especially in summarizing key findings.
  • Some sentences are incomplete or grammatically incorrect.
  • The key takeaway message is not clearly defined.

Specific Recommendations:

  1. Clarify the main objective.
  • Current wording:
  • “Here we, gait and health. However, footwear selection rarely takes this into account.” (lines 13-15) – This sentence is incomplete and unclear.
  • Suggested revision:
  • “Here, we examine the impact of different types of footwear on gait and health. However, footwear selection rarely considers these biomechanical implications.”

*** Response 1 ***

We thank the reviewer not just for their comments but especially for making specific recommendations. This is very helpful. We have followed these recommendations, and believe the text has no grammatical errors.

*** Comment 2 ***

  1. Improve sentence structure and clarity.
  • Current wording:
  • “Our review suggest that minimal footwear, which interacts minimally with the way we walk, is preferable for healthy peopleto stimulate long-term health in the general pop-ulation.” (lines 17-19) – Contains grammatical errors.
  • Suggested revision:
  • “Our review suggests that minimal footwear, which minimally interferes with natural gait, may be preferable for promoting long-term musculoskeletal health in the general population.”

*** Response 2 ***

Thank you, we have followed this suggestion.

*** Comment 3 ***

  1. Abstract should emphasize findings rather than general statements.
  • Consider adding specific findings from the review, such as the effect of footwear on muscle activation, joint loading, and foot morphology.

*** Response 3 ***

We believe the Abstract is as specific as possible, recommending minimal footwear. However we gave not gone in detail about muscle activation and joint loading, since we do not deal with this in detail in the paper.

*** Comment 4 ***

  1. Introduction

Issues:

  • The introduction is well-grounded in evolutionary context, but it does not clearly articulate the research gap.
  • Some sentences are redundant or overly complex.
  • The transition from barefoot walking history to modern footwear issues is abrupt.

Specific Recommendations:

  1. Improve clarity in the opening statement.
  • Current wording:
  • “Humans are the only bipedal primates and have been bipedal for up to 6-7 million years [1]. This means that our feet are the only mechanical interface with the substrate.” (lines 32-34)
  • Suggested revision:
  • “Humans are the only fully bipedal primates, relying on their feet as the primary mechanical interface with the ground. Bipedalism has shaped the evolutionary development of the foot’s structure and function.”

*** Response 4 ***

Thank you, we have amended our Introduction based on these suggestions.

*** Comment 5 ***

  1. Strengthen the transition between historical context and modern footwear concerns.
  • Current wording:
  • “Because a shoe sits physically between the foot and the substrate, it will affect the biomechanics the foot and of overall gait to some degree.” (lines 40-41) – awkward phrasing.
  • Suggested revision:
  • “Modern footwear serves as an interface between the foot and the environment, altering foot biomechanics and potentially influencing gait patterns.”

*** Response 5 ***

Thank you, we have reworded based on this suggestion and made the transition between historical and modern shoes clearer.

*** Comment 6 ***

  1. Clarify research gap and objectives.
  • The manuscript should explicitly state why this review is necessary.
  • Suggested addition:
  • “Despite the prevalence of conventional footwear, its long-term biomechanical effects remain poorly understood. This review aims to evaluate current evidence on how different types of footwear influence lifelong locomotor health.”

*** Response 6 ***

Thank you – suggestion followed although not verbatim, in order to respect the flow of the Introduction.

*** Comment 7 ***

  1. Methods

Issues:

  • Lack of systematic methodology. The paper does not clearly outline inclusion/exclusion criteria for studies.
  • No clear justification for database selection (Google Scholar and PubMed).
  • Potential selection bias as the methodology is not transparent.

Specific Recommendations:

  1. Define a structured search strategy.
  • Suggested revision (to be added in the methods section):
  • “A literature search was conducted using PubMed and Google Scholar up to November 2024. Keywords included ‘footwear biomechanics,’ ‘barefoot walking,’ ‘foot morphology,’ and ‘gait analysis.’ Only peer-reviewed English-language studies were included.”

*** Response 7 ***

Thank you. We have further elaborated on the type of review this is, and on our search strategy. As a critical/narrative review it does not have the very strict requirements (and limitations) of e.g. a Systematic Review. As a response to another reviewer (Reviewer 3), we have consulted the SALSA methodology (and partly adopted it as it is specific to systematic reviews) and the very useful Grant and Booth (2009, Health Inf Libr J, 26:91-108) paper, which states on the topic of a critical review that “there is no formal requirement to present methods of the search, synthesis and analysis explicitly”. However we have provided some extra detail where possible.

*** Comment 8 ***

Explain study selection criteria.

  • The manuscript should state:
  • “We included studies comparing habitually barefoot and shod populations, experimental studies on footwear effects, and epidemiological research on foot health.”

*** Response 8 ***

Thank you, we have added this.

*** Comment 9 ***

  1. Clarify how the review was conducted.
  • Was it a narrative review or a semi-systematic review? The authors should specify their approach.

*** Response 9 ***

This is a critical review, one type of narrative review. We have elaborated substantially on this topic in a previous round of reviews. Based on Sukhera (2022), we have termed the paper as a Critical Review, one specific subtype of Narrative Review, and this is described in the Methods section.

*** Comment 10 ***

  1. Results & Discussion

Issues:

  • Overgeneralized conclusions without sufficient nuance.
  • Limited discussion of confounding factors (e.g., socioeconomic status, surface hardness, cultural differences).
  • Some claims lack strong supporting evidence.

Specific Recommendations:

  1. Avoid absolute statements when evidence is mixed.
  • Current wording:
  • “Conventional footwear lacks biomechanical benefits.” (lines 495-496)
  • Suggested revision:
  • “While some design elements of conventional footwear, such as excessive cushioning and arch support, may alter natural foot mechanics, their long-term impact on health remains debated.”

*** Response 10 ***

Thank you. We have rephrased some of the sentences to be elaborate on other confounders, and are more cautious, where appropriate. The phrase on lines 495-496 was not included in the last version of the manuscript.

*** Comment 11 ***

Include discussion on adaptation to footwear.

  • Some studies suggest that foot adaptations to footwear occur over time.
  • Suggested addition:
  • “Recent studies indicate that habitual use of footwear may lead to structural and functional adaptations in the foot, which should be considered when evaluating the effects of conventional vs. minimal footwear.”

*** Response 11 ***

Thank you. For the sake of brevity, we have chosen to keep the original phrase, but we hope it is clear from the manuscript as a whole that footwear can indeed lead to structural and functional adaptations in the foot.

*** Comment 12 ***

Acknowledge potential benefits of certain shoe features.

  • Raised heels can be useful in conditions such as Achilles tendinopathy.
  • Arch support may benefit individuals with flat feet experiencing pain.

*** Response 12 ***

Thank you, we have acknowledged those potential benefits for the named pathological conditions.

*** Comment 13 ***

Conclusion

Issues:

  • The conclusion is too strong given the available evidence.
  • The call to action lacks nuance—suggesting that all conventional footwear is harmful is an overgeneralization.

Specific Recommendations:

  1. Rephrase the concluding statement to reflect a more balanced perspective.
  • Current wording:
  • “Minimal footwear should be the default choice for the general population.” (lines 500-501)
  • Suggested revision:
  • “Based on current evidence, minimal footwear may offer biomechanical advantages for healthy individuals. However, specific footwear choices should consider individual needs, environmental conditions, and potential medical requirements.”

*** Response 13 ***

We have added nuance as suggested; in fact this was already the case in the latest version of the manuscript.

*** Comment 14 ***

  1. Highlight knowledge gaps for future research.
  • Suggested addition:
  • “Further research is needed to determine the long-term effects of different footwear types across diverse populations and varying environmental conditions.”

*** Response 14 ***

Thank you, this is a good suggestion and we have amended the manuscript accordingly.

*** Comment 15 ***

Final Remarks

  • The manuscript covers an important and underexplored topic, but improvements in clarity, methodology, and balance of argumentation are necessary.
  • A clearer methodological framework and more nuanced discussion would greatly enhance its scientific impact.
  • Language polishing is recommended to improve readability and coherence.

*** Response 15 ***

Thank you, with the suggestions you and the other reviewers made, we believe it has now further improved.

*** Comment 16 ***

Comments on the Quality of English Language

The manuscript is generally understandable; however, it contains numerous grammatical errors, awkward phrasing, and inconsistent terminology. Some sentences are unclear or incomplete, affecting readability. Additionally, word choice and sentence structure could be improved for better clarity and fluency.

*** Response 16 ***

Thank you, we believe our manuscript has now further improved in this regard.

Reviewer 3 Report (New Reviewer)

Comments and Suggestions for Authors

I would like to thank the authors for submitting their article, which provides an insight into the relationship between footwear and locomotor health.

After running the article through Turnit in, an anti-plagiarism software for cross-checking, the first result is 45%. After excluding quotes, bibliography and matches that are less than 1%, the match is 4%, which speaks in favour of the originality of the work.

Reviewer:

The title of the article is sufficient, clear and precise.

Recommendation/ Suggestion

Footwear choice and locomotor health throughout the life course: a Literature Review

Please consult the literature and find the best possible title that describes your research methodology.

Why Literature review:

Description: Generic term: published materials that provide examination of recent or current literature. Can cover wide range of subjects at various levels of completeness and comprehensiveness. May include research findings

Search: May or may not include comprehensive searching

Synthesis: Typically narrative

 Analysis: may be chronological, conceptual, thematic, etc.

Consult the SALSA methodology

Reviewer:

The abstract contains about 233 words with simple summary.

Reviewer:

Arrange the keywords in the correct order. Keywords are not precise. Please correct.

1. Introduction

Reviewer:

The introduction to the article is clear and contains a sufficient number of quotations. It introduces the reader to the purpose of the article and to the problem.

2. Methods

The methodology is clearly formulated. Methodology is sufficiently described.

Kindly ask to remove lines from 96 to 98.

Unnecessary explanation of the division of the article. The titles are sufficient.

3. Foot health and footwear/ 3.1 Long-term effects of footwear: comparing habitually barefoot and habitually shod populations/3.2 Long-term effects of footwear: foot development in children/3.3  Short term effects of footwear: studies on barefoot, minimally shod and conventionally shod walking in habitually shod populations

This section's of article are clear and contains a sufficient number of relevant quotations. No objections

4. Effects of specific footwear features on the structure and function of the foot/ 4.1 Raised heel/ 4.2 Cushioning/ 4.3 Arch support/4.4  Last shape and shoe size/4.5 Bending and torsional stiffness/ 4.6 Rigid heel counter

Reviewer:

This section's of article are clear and contains a sufficient number of relevant quotations. No objections

Recommendation/ Suggestion

I suggest that you divide this article into an introduction, methodology, discussion with subheadings (e.g 3. Discussion than follows 3.1 subheadings, 3.2 subheadings, 3.3 etc til last 4.6 subheadings) and conclusion. Then the article would be more systematic and meaningful.

5. Conclusion

Reviewer.

The authors draw a satisfactory conclusion after the reviewing the literature.

6. Literature

Reviewer:

The literature is not cited according to the journal's guidelines. Please make corrections. Missing doi.

Please make correction where is possible, regarding the doi.

Author Response

Reviewer 3

*** Comment 1 ***

I would like to thank the authors for submitting their article, which provides an insight into the relationship between footwear and locomotor health.

After running the article through Turnit in, an anti-plagiarism software for cross-checking, the first result is 45%. After excluding quotes, bibliography and matches that are less than 1%, the match is 4%, which speaks in favour of the originality of the work.

*** Response 1 ***

Thank you. We can confirm that this manuscript is entirely our writing, and would also like to add for complete transparency that we have not used generative AI to preserve academic integrity.

*** Comment 2 ***

Reviewer:

The title of the article is sufficient, clear and precise.

Recommendation/ Suggestion

Footwear choice and locomotor health throughout the life course: a Literature Review

Please consult the literature and find the best possible title that describes your research methodology.

*** Response 2 ***

Thank you. We believe that our title, which has changed in previous round of reviews, covers the contents of the paper; we state a “critical” review rather than “literature” review since it is more specific and has resulted from previous reviews.

*** Comment 3 ***

Why Literature review:

Description: Generic term: published materials that provide examination of recent or current literature. Can cover wide range of subjects at various levels of completeness and comprehensiveness. May include research findings

Search: May or may not include comprehensive searching

Synthesis: Typically narrative

Analysis: may be chronological, conceptual, thematic, etc.

Consult the SALSA methodology

*** Response 3 ***

Thank you, we have consulted the SALSA methodology (which applies most systamatic reviews) and the very useful paper by Grant and Booth (2009, Health Inf Libr J, 26:91-108), which states that “here is no formal requirement to present methods of the search, synthesis and analysis explicitly”. However, we have added detail as appropriate.

*** Comment 4 ***

Reviewer:

The abstract contains about 233 words with simple summary.

Reviewer:

Arrange the keywords in the correct order. Keywords are not precise. Please correct.

*** Response 4 ***

Keywords are now in alphabetical order.

*** Comment 5 ***

  1. Introduction

Reviewer:

The introduction to the article is clear and contains a sufficient number of quotations. It introduces the reader to the purpose of the article and to the problem.

  1. Methods

The methodology is clearly formulated. Methodology is sufficiently described.

*** Response 5 ***

Thank you.

*** Comment 6 ***

Unnecessary explanation of the division of the article. The titles are sufficient.

*** Response 6 ***

We also feel the paper does not strictly need this phrase, but this was put in upon the request of several other reviewers, so we believe it may help for some readers too.

*** Comment 7 ***

  1. Foot health and footwear/ 3.1 Long-term effects of footwear: comparing habitually barefoot and habitually shod populations/3.2 Long-term effects of footwear: foot development in children/3.3 Short term effects of footwear: studies on barefoot, minimally shod and conventionally shod walking in habitually shod populations

This section's of article are clear and contains a sufficient number of relevant quotations. No objections

  1. Effects of specific footwear features on the structure and function of the foot/ 4.1 Raised heel/ 4.2 Cushioning/ 4.3 Arch support/4.4 Last shape and shoe size/4.5 Bending and torsional stiffness/ 4.6 Rigid heel counter

Reviewer:

This section's of article are clear and contains a sufficient number of relevant quotations. No objections

Recommendation/ Suggestion

I suggest that you divide this article into an introduction, methodology, discussion with subheadings (e.g 3. Discussion than follows 3.1 subheadings, 3.2 subheadings, 3.3 etc til last 4.6 subheadings) and conclusion. Then the article would be more systematic and meaningful.

*** Response 7 ***

Thank you, our manuscript has had many structural changes in a previous round of reviews and changing the current structure may negate other reviewer’s comments. We hope the current structure is sufficiently clear, and feel that a separate Discussions section is more suitable for papers with novel experimental data.

*** Comment 8 ***

  1. Conclusion

Reviewer.

The authors draw a satisfactory conclusion after the reviewing the literature.

  1. Literature

Reviewer:

The literature is not cited according to the journal's guidelines. Please make corrections. Missing doi.

Please make correction where is possible, regarding the doi.

*** Response 8 ***

The journal does not require inclusion of a DOI, however this is a good idea and we have consulted with the editor. They have confirmed that we can keep the reference list as is and that useful changes will be made during the production of the print proofs.

This manuscript is a resubmission of an earlier submission. The following is a list of the peer review reports and author responses from that submission.

Round 1

Reviewer 1 Report

Comments and Suggestions for Authors

This article is not scientifically written.

For example, in the conclusion, there is no need for reference. and is too long.

The contents are stated irregularly and without proper sequence.

The conclusion is biased.

The articles presented in the tables are not representative of all the articles published in this field.

The method is unclear.

The articles used are biased.

Author Response

Reviewer 1

General response:

We thank the reviewer for their comments. We have made numerous changes upon their (and the other reviewers) comments. In some cases we found the comment not very clear, and as a result, we found it difficult to amend our manuscript accordingly.

Comment 1:

This article is not scientifically written.

For example, in the conclusion, there is no need for reference. and is too long.

Response:

Thank you, we have removed the reference from the conclusion and shortened it. We regret the reviewer feels our paper is not scientifically written. I have not had this comment before for any of my peer-reviewed publications, so I would benefit from more specifics to understand what exactly is meant by the reviewer.

Comment 2:

The contents are stated irregularly and without proper sequence.

Response:

Thank you, we have worked on the structure of the paper and believe it now has a better logic and flow.

Comment 3:

The conclusion is biased.

Response:

The reviewer has not clarified in what sense they feel our conclusions are biased, so it is difficult to respond to this comment. However, one our mains aims is to stimulate a discussion about “what is healthy footwear”, and if our conclusion does that we have achieved our goal.

Comment 3:

The articles presented in the tables are not representative of all the articles published in this field.

Response:

Thank you, and we have now removed the tables. We have also simplified the figure.

Comment 4:

The method is unclear.

Response:

We have clarified our method in the revised version. This paper is a narrative review and follows the principles and methods of a narrative review (Sukhera, 2022, J Grad Med Educ 14:414-417).

Comment 5:

The articles used are biased.

Response:

The reviewer has not clarified in what sense they feel the references are biased, so it is difficult to respond to this comment.

Reviewer 2 Report

Comments and Suggestions for Authors

Thank you for your work. I have completed reading this narrative review on the effect of different types of footwear on kinematics, kinetics, and health. I think the study's significance is not well established. Further, the scope of the study should be better highlighted. So, it may be difficult for the readers to understand its significant contribution to scientific literature and clinical practice.

Author Response

Reviewer 2

General response:

We thank this reviewer for their insightful comments. We especially appreciate that they have focussed on the bigger picture (esp. the scope of the paper) and we agree with their comments. We feel the paper has much benefited from their comments in terms of scope, clarity and conciseness.

Comment 1:

Thank you for your work. I have completed reading this narrative review on the effect of different types of footwear on kinematics, kinetics, and health. I think the study's significance is not well established. Further, the scope of the study should be better highlighted. So, it may be difficult for the readers to understand its significant contribution to scientific literature and clinical practice.

Response:

Thank you. In response to this comment, and that of another reviewer, we have worked on the structure of the paper as well as made its scope and significance clearer from the onset (i.e. at the end of the Introduction). We hope the reviewer is satisfied with our amendments. Our paper aligns strongest with the Journal’s published formal scope of “Prevention”.

More specifically, our scope is the influence of daily footwear on lifelong (locomotor) health and we have now state this clearly at the end of the Introduction. Our scope does not include work on patients with locomotor issues and for which footwear (incl. orthotics) fulfils a very specific treatment role. It also does not include sports footwear which similarly has functions beyond lifelong health, such as improving athletic performance, or avoiding sport-specific injuries. We have made this clear but we have cited some clinical and sports oriented work where relevant.

Comment 2:

The title should be more descriptive. Further I cannot understand research design described by the words “Position Paper” in “A Narrative Review and Position Paper”

Response:

Thank you and we agree we have been unclear. We have rewritten the title and hope it is much more suitable now.

Comment 3:

I think the paper needs to be thoroughly read and proof-read for clarity, presentation, grammar, subject-verb agreement, use of English, etc.

Response:

Thank you, we have had another round of proof-reading (incl. by native speaker) and made changes throughout the manuscript.

Comment 4:

I think the presentation throughout the paper should be more concise. Specific concerns: Abstract:

Please highlight the importance of your research and better specify its scope.

Response:

Thank you, we have shortened and clarified the abstract and made chances throughout the manuscript to make it more concise.

Comment 5:

Introduction:

Please provide better justifications of your research.

Response:

We have now made our aims and scope clearer (see end of the Introduction). We feel the justification for our research is now much clearer. In a nutshell, we argue that minimal footwear is the best default option to promote life-long health, and deviations from this should be underpinned be evidence as we now put clearly in our, more brief, Conclusion..

Comment 6:

Please highlight what is specifically the scope of your study. You report at the abstract that you studied “effects of daily footwear on kinematics, kinetics and health”. Please highlight what is meant by “kinematics, kinetics, and health” which variables would be considered to reflect each of them? Why?

Response:

Thank you, we agree this was not clearly worded and have amended. We have removed the unnecessary and confusing reference to “kinematics, kinetics and health” which made sense in  a previous version of the manuscript but not in the current one.

Comment 7:

Effects of ageing on the foot in general and on the heel pad in particular

Please better relate this section to the scope of your study

Response:

Thank you and this is an excellent comment. During the writing process, we have added this section later and regret that it does not relate to the rest clearly enough. We have now rewritten to integrate this section better with the rest of the manuscript. The original, separate section has gone and its contents have been incorporated elsewhere. For example, the details on the heel pad have now been incorporated with the section on “cushioning” (which is now section 3.2).

Reviewer 3 Report

Comments and Suggestions for Authors

To the author

Congratulations on a great paper.

Here are my comments on the paper.

There is not enough information about the research methods. For example, you don't describe which academic databases you used, what keywords you used to search, how long you searched for articles, and how many articles were selected from the total number of articles.

I think it would be a good paper if you added these things.

Have a good day.

Author Response

Reviewer 3

Comment 1:

Congratulations on a great paper.

Here are my comments on the paper.

There is not enough information about the research methods. For example, you don't describe which academic databases you used, what keywords you used to search, how long you searched for articles, and how many articles were selected from the total number of articles.

I think it would be a good paper if you added these things.

Have a good day.

Response:

Thank you for your positive evaluation of our manuscript. We have now added some details about our methods. Because this is an important and justified comment, please allow us to elaborate somewhat on our approach and, following from this, our methods.

This is not an experimental research paper – rather we propose a new working hypothesis around footwear choice and substantiate this hypothesis with current evidence. Our hypothesis is that minimal shoes are the best starting point to promote long-term health, and that deviations from this (i.e. adding “non minimal” features such as a raised heel) are best only implemented if there is evidence in favour of them. We have now explained this better – the term “position paper” itself is probably unclear indeed.

We have chosen to tackle this subject as a “narrative review”. We felt this approach was particularly suitable, since we wanted to describe what is known about the very broad topic of footwear and provide a new way of thinking, making a narrative review particularly suitable (as explained in Sukhera, 2022, J Grad Med Educ 14:414-417). Narrative reviews typically are flexible and do not involve strict in- and exclusion criteria allowing for a practical synthesis of a topic (Sukhera, 2022) and find an overall background for a specific issue (Demiris et al., 2019, In: Behah Interv Res Hospice Palliat Care). We have used PubMed and Google scholar (and added a few references incl. from 2024). We selected references to align with the scope of our study, while aiming to include alternative views. A systematic review would not be suitable in our case; they lend themselves more to very tightly defined subjects with a narrow scope, and we are considering this for a future systematic review paper focussing exclusively on heel impact cushioning.

Reviewer 4 Report

Comments and Suggestions for Authors

I congratulate the authors on their submission, which explores the health effects of footwear usage. The topic is interesting and addresses important questions. However, the manuscript does not follow the structured approach typically expected in scientific publications. Specifically, it lacks a transparent methodology and structured sections for results, discussion, and conclusions.

The manuscript begins without an introduction to the paper's structure or a clear overview of the arguments. As a result, the rationale behind the content and the sequence of chapters is not immediately clear.

For example, it is unclear why the section on age-related changes precedes the anatomical discussion of the foot. If age-related changes are to be discussed, it would be more logical to include both juvenile and elderly modifications together. It would be advisable to address the health impacts of footwear usage in normal adult life and then move on to the childhood and elderly stages.

While the paper mentions genetic and learned influences on foot shape during walking, these factors are not presented in the comparative evaluation of the studies cited. Additionally, the role of foot size and proportional dimensions should be clarified when discussing foot anatomy, particularly in relation to the shoe's upper part.

The manuscript mentions several methods, for example, in lines 115 to 131, but it is not comprehensive. For example, it does not mention pressure distribution analysis or measurements of contact forces during walking.

As I read the paper, I understood that the health consequences arise more from walking, running, or jumping than from wearing footwear. This raises the question of why the act of wearing footwear is emphasized over the dynamics of movement while wearing shoes.

Overall, it is crucial to clearly define the paper's structure, identify the argument's key elements, and base the review on a narrower but more systematically chosen literature. For example, the paper could first introduce foot anatomy, then describe the functions of different types of footwear, detailing the corresponding characteristics of footwear parts, followed by an objective review of the positive and negative effects based on the literature.

It would also be beneficial to include a practical section with evidence-based recommendations for proper footwear selection and usage. This could address aspects such as comfort, health, satisfaction, and specific issues like shoelace tightness or heel height.

In line 74, it is stated that this is a non-exhaustive review, which is questionable considering the reference list contains 178 items.

Additionally, it is unclear what kind of impact is being referred to in line 357.

Comments on the Quality of English Language

In line 456, it is not appropriate to refer to a "summary," as this is the role of the conclusions section. This phrasing suggests that a large-language model might have been used in drafting the article.

The introductory section contains overly complex sentences, with several insertions in parentheses that make reading more difficult. The closing parenthesis is also missing from line 57.

Author Response

Reviewer 4

General response:

We would like to thank the reviewer for their constructive and justified comments. We really feel they have greatly helped improve the paper. Their general comments about structure and methods is also reflected by other reviewers, so we have tried to improve especially in that respect.

Comment 1:

I congratulate the authors on their submission, which explores the health effects of footwear usage. The topic is interesting and addresses important questions. However, the manuscript does not follow the structured approach typically expected in scientific publications. Specifically, it lacks a transparent methodology and structured sections for results, discussion, and conclusions.

The manuscript begins without an introduction to the paper's structure or a clear overview of the arguments. As a result, the rationale behind the content and the sequence of chapters is not immediately clear.

For example, it is unclear why the section on age-related changes precedes the anatomical discussion of the foot. If age-related changes are to be discussed, it would be more logical to include both juvenile and elderly modifications together. It would be advisable to address the health impacts of footwear usage in normal adult life and then move on to the childhood and elderly stages.

Response:

We have worked on the structure and hope the paper now has a much better intent and flow. Specifically on the ageing section – this was also (correctly) highlighted by Reviewer 2. This section is now gone as a separate entity, and its contents are integrated with the rest of the manuscript (e.g. the section of heel pads is now integrated with section 3.2 on Cushioning). For clarity, we have uploaded a tracked-changes version of the manuscript, in addition to the typeset version the journal requires.

Comment 2:

While the paper mentions genetic and learned influences on foot shape during walking, these factors are not presented in the comparative evaluation of the studies cited. Additionally, the role of foot size and proportional dimensions should be clarified when discussing foot anatomy, particularly in relation to the shoe's upper part.

Response:

Thank you for this comment. Both genetic and developmental (incl. phenotypic plasticity) aspects play a role (see e.g. ethnic/genetic differences in foot shape: Kouchi, 1998, Anthropol Sci). However, only the latter can be influenced by footwear and is thus the focus of our paper. We have added a phrase at the end of (what is now) section 2.1 stating that foot shape and size can differ between populations (but this is not our focus) and other factors such as sex, and we have also clarified this (and referenced a recent systematic review on the topic; Hoey et al, 2022, Footwear Sci) in section 3.4 on last shape and foot size.

Comment 3:

The manuscript mentions several methods, for example, in lines 115 to 131, but it is not comprehensive. For example, it does not mention pressure distribution analysis or measurements of contact forces during walking.

Response:

We agree that we do not provide a comprehensive overview of methods. We feel that this would make the paper heavier than it needs to be so we have focussed on the most relevant ones for the topic at hand. For the future, we are considering a focussed systematic review on one aspect only (cushioning) which will incorporate all methods.

Comment 4:

As I read the paper, I understood that the health consequences arise more from walking, running, or jumping than from wearing footwear. This raises the question of why the act of wearing footwear is emphasized over the dynamics of movement while wearing shoes.

Response:

The health effects of physical activity (like walking  running or jumping) and footwear choice might be difficult to fully disentangle in some cases, because (for example) more barefoot populations might also be more active. However, where 1-to-1 comparisons are possible (similar physical activity) we still see an effect, e.g. in the studies on African children or our own work from India (on barefoot walkers) and from the UK (on foot strength) referenced in the paper. Moreover, footwear, not physical activity, is our scope. The benefits of physical activity on health have been well documented in the literature.

Comment 5:

Overall, it is crucial to clearly define the paper's structure, identify the argument's key elements, and base the review on a narrower but more systematically chosen literature. For example, the paper could first introduce foot anatomy, then describe the functions of different types of footwear, detailing the corresponding characteristics of footwear parts, followed by an objective review of the positive and negative effects based on the literature.

Response:

Thank you and we agree with this concern. We have worked on its structure and made it generally much clearer (incl. the description of footwear characteristics and their effect on the foot, gait and health).

In terms of the literature chosen, we found the “narrative review” approach the most suitable, since we wanted to describe what is known about the very broad topic of footwear and provide a new way of thinking. A narrative review is particularly suitable for this (as explained in Sukhera, 2022, J Grad Med Educ 14:414-417). Narrative reviews typically are flexible and do not involve strict in- and exclusion criteria allowing for a practical synthesis of a topic (Sukhera, 2022) and find an overall background for a specific issue (Demiris et al., 2019, In: Behah Interv Res Hospice Palliat Care). We selected references to align with the scope of our study, while aiming to include alternative views. A systematic review would not be suitable in our case; they lend themselves more to very tightly defined subjects with a narrow scope, and we are considering this for a future systematic review paper focussing exclusively on heel impact cushioning.

We feel our manuscript aligns well with Healthcare’s mission: “Healthcare hopes to influence global health and disease aspects (…)” (https://www.mdpi.com/journal/healthcare/healthcare_flyer.pdf)

We strongly believe (and provide underpinning evidence) that choices as simple as “which shoes shall I wear” can make a big impact on long-term health. Even if our working hypothesis did prove wrong, we would see this as a positive outcome, because there would have been  a debate. It would mean footwear has become a deliberate health choice, and not a non-choice (or a pure fashion choice) as is often the case now.

Comment 6:

It would also be beneficial to include a practical section with evidence-based recommendations for proper footwear selection and usage. This could address aspects such as comfort, health, satisfaction, and specific issues like shoelace tightness or heel height.

Response:

Thank you and I agree this would be the ultimate goal, but we feel we need more future work to make such recommendations with confidence. Footwear choice is also going to be very personal, and we believe bespoke, 3D printed footwear might become common. At this time, we recommend to explore minimal footwear (with care), and have added a statement to that effect including in the Abstract.

Comment 7:

In line 74, it is stated that this is a non-exhaustive review, which is questionable considering the reference list contains 178 items.

Response:

Thank you, and we agree it is a long list of references but we have addressed many aspects in our paper and did not perform a systematic review for reasons outlined higher. With the re-write of the introduction and a new statement (and a few references) on the methods of a narrative review, the term “non-exhaustive” has become redundant and has been removed.

Comment 8:

Additionally, it is unclear what kind of impact is being referred to in line 357.

Response:

We mean the initial foot contact phase, i.e. when the foot impact with the substrate. We have rewritten this sentence to be much clearer.

Comment 9:

Comments on the Quality of English Language

In line 456, it is not appropriate to refer to a "summary," as this is the role of the conclusions section. This phrasing suggests that a large-language model might have been used in drafting the article.

Response:

We believe this refers to the statement “In summary, conventional western shoes…” (line 459 in our version). We agree and have reworded to remove the “Summary” here and integrate with the conclusion.

However, we would like to stress that we have not used any language model/ generative AI (ChatPGT or other) to generate any text in our manuscript. Personally, I (KD) would like to state that I have major ethical and scientific objections against the use of Generative AI in scientific writing.

Comment 10:

The introductory section contains overly complex sentences, with several insertions in parentheses that make reading more difficult. The closing parenthesis is also missing from line 57.

Response:

Thank you, we have now improved our writing, corrected typographical errors and favoured shorter sentences.

Round 2

Reviewer 1 Report

Comments and Suggestions for Authors

Since the article is a narrative review, it has bias and its results are not reliable.

It would have been better to conduct a systematic review.

Given that the article is a narrative review, the previous drawbacks still exist.

Finally, a decision regarding the article should be made based on the opinion of the respected editor.

Author Response

Reviewer 1, round 2

Comment:

Since the article is a narrative review, it has bias and its results are not reliable.

It would have been better to conduct a systematic review.

Given that the article is a narrative review, the previous drawbacks still exist.

Finally, a decision regarding the article should be made based on the opinion of the respected editor.

Response:

Thank you. Indeed, this paper is a narrative review and hence does not have the focus and rigour of a systematic review. However, we believe a narrative review is the most appropriate type of publication in our case, for the reasons outlined previously and mentioned in the references we provided. In a nutshell, we believe our broad scope is more suited to a narrative review, and this will also help stimulate the debate on healthy footwear that we believe is needed. Going forward, we are considering a paper on one very specific topic, i.e. heel cushioning during walking, and in this case a systematic review will be the most appropriate publication type.

Reviewer 4 Report

Comments and Suggestions for Authors

Thank you for the improved version.

Comments on the Quality of English Language

You should change sex to gender.

Author Response

Reviewer 4, round 2

Comment:

Thank you for the improved version.

You should change sex to gender.

Response:

Thank you, and we really appreciate this comment on a complex but important issue (scientifically and societally). We find it important not to conflate sex, a biological concept (in humans: male, female, intersex), and gender, a psychosocial concept (man, woman, transgender, non-binary etc.) (see also Donovan et al., 2024, Science 383(6685): 822-825).

We have carefully checked what the intended meaning was in the three instances where we refer to “sex”.

The first reference to “sex” (line 274) does not specify (as expected from an old paper). In line with the reviewer’s comment, we find it appropriate to change our wording from “all sexes” (which is technically correct because it includes everyone) to “all genders” which we feel is also correct, but more inclusive.

The second reference to “sex” (line 312) is about anatomical/biological differences in the heel pad, and we have therefore kept the term “sex”; in addition this is the term used by the authors (Ugbolue et al., 2020, J Anat).

The third reference (line 424) is explicitly about “sex” and the authors have defined its use: “the term sex was used as it is biological” (Hoey et al, 2023, Footwear Sci, p. 56). Therefore we have kept the term.

Please note that we have used the gender term “women” in line 409 (as per the original paper) and “gender” in line 410.

Going forward, we encourage future research to explicitly define whether they address sex or gender (or both) as appropriate.